# Warming Scenarios and *Phytophthora cinnamomi* Infection in Chestnut (*Castanea sativa* Mill.)

**DOI:** 10.3390/plants12030556

**Published:** 2023-01-26

**Authors:** F. Javier Dorado, Juan Carlos Alías, Natividad Chaves, Alejandro Solla

**Affiliations:** 1Faculty of Forestry, Institute for Dehesa Research (INDEHESA), Avenida Virgen del Puerto 2, Universidad de Extremadura, 10600 Plasencia, Spain; 2Department of Plant Biology, Ecology and Earth Sciences, Faculty of Science, Universidad de Extremadura, 06080 Badajoz, Spain

**Keywords:** abiotic stress, climate change, biochemistry

## Abstract

The main threats to chestnut in Europe are climate change and emerging pathogens. Although many works have separately addressed the impacts on chestnut of elevated temperatures and *Phytophthora cinnamomi* Rands (*Pc*) infection, none have studied their combined effect. The objectives of this work were to describe the physiology, secondary metabolism and survival of 6-month-old *C. sativa* seedlings after plants were exposed to ambient temperature, high ambient temperature and heat wave events, and subsequent infection by *Pc*. Ten days after the warming scenarios, the biochemistry of plant leaves and roots was quantified and the recovery effect assessed. Plant growth and root biomass under high ambient temperature were significantly higher than in plants under ambient temperature and heat wave event. Seven secondary metabolite compounds in leaves and three in roots were altered significantly with temperature. Phenolic compounds typically decreased in response to increased temperature, whereas ellagic acid in roots was significantly more abundant in plants exposed to ambient and high ambient temperature than in plants subjected to heat waves. At recovery, leaf procyanidin and catechin remained downregulated in plants exposed to high ambient temperature. Mortality by *Pc* was fastest and highest in plants exposed to ambient temperature and lowest in plants under high ambient temperature. Changes in the secondary metabolite profile of plants in response to *Pc* were dependent on the warming scenarios plants were exposed to, with five compounds in leaves and three in roots showing a significant ‘warming scenario’ × ‘*Pc*’ interaction. The group of trees that best survived *Pc* infection was characterised by increased quercetin 3-O-glucuronide, 3-feruloylquinic acid, gallic acid ethyl ester and ellagic acid. To the best of our knowledge, this is the first study addressing the combined effects of global warming and *Pc* infection in chestnut.

## 1. Introduction

Plants are exposed to a multitude of stresses that often occur simultaneously [1,2,3]. Plant response mechanisms to an individual stressor often differ from those to combined stress, mostly because of complex synergistic or antagonistic interactions occurring between hosts and stressors [4,5,6,7]. To easily identify plant response mechanisms to combined stress, preliminary studies should be performed under controlled conditions [8,9]. Regardless of the type of stress or the combination, plants accumulate secondary metabolites in response to stress, in some cases allowing plants to mitigate damage [10,11]. Secondary metabolites are a diverse group of compounds of low molecular weight, mostly synthesised from products of primary carbon metabolism. Temperature has been shown to alter plant secondary metabolism, and this response seems to be species- compound- or context-dependent [12]. Compounds from the secondary metabolism of plants may directly interact with pathogens (e.g., as antifungal agents) [13,14] or participate in the immune response of plants [15].

Chestnut (*Castanea sativa* Mill.) is a Fagaceae tree species of considerable economic and environmental significance, occurring in the Mediterranean region. Climate models predict that this region will undergo temperature increase, heat wave events and extreme drought [16]. Studies on the impact of climate change on chestnut, including tolerance, adaptation and genetic variability, have increased considerably in recent years [17,18,19,20,21]. Because this species developed in contrasting climate conditions, different evolutionary pressures acted on its genome, giving rise to ecotypes adapted to different climates [18,22,23,24]. The genetic variability, physiology and biochemistry of chestnut in response to heat and drought stress have been studied [24,25,26,27,28,29], providing knowledge for breeders to obtain plant material resilient to the new conditions of climate change. However, little information is available on the response of *C. sativa* to combined abiotic and biotic stress.

Ink disease, caused mainly by *Phytophthora cinnamomi* Rands, is considered the most widespread and destructive disease of *C. sativa* [30,31,32,33]. Symptoms include fine root rot, root collar necrosis, obstruction of xylem vessels, leaf chlorosis, rapid or gradual wilting of leaves and dieback [14,33,34]. *Phytophthora cinnamomi* (*Pc*) is a soil-borne pathogen (Stramenopila, Oomycota) that infects almost 5000 plant species, including many that are significant in agriculture, forestry and horticulture [30]. Recent studies on chestnut have addressed the genetic background, adaptation, maternal effects, regeneration and histology of trees in response to *Pc* infection [35,36,37,38,39,40] and elucidated the metabolic, proteomic and hormonal changes in chestnut [10,41,42]. The impact of ink disease on chestnut depends on the environment, and is more severe in areas with high temperature, low relative humidity and dry summers [43]. *Phytophthora cinnamoni* appears to have a large impact on species from the Mediterranean-type climate, where mild, wet conditions in autumn and spring, ideal for sporulation and host infection, alternate with hot, dry summers that are unfavourable for plants [44]. A combination of abiotic and biotic factors is thought to be behind the decline of oak, jarrah, beech and pine trees [29,45,46,47]. Studies have addressed the impact of *Pc* diseases in combination with abiotic stress, such as drought, in several tree species [9,48,49,50]. However, in *C. sativa*, the combined effect of abiotic and *Pc* stress has not been studied. Global warming has created an urgent need to study whether temperature influences the physiology, biochemistry and survival of chestnut in response to *Pc*. The objectives of this work were to test the following hypotheses: (i) high ambient temperature and heat wave events induce physiological and biochemical changes in *C. sativa*, (ii) infection of *C. sativa* seedlings by *Pc* increases the abundance of secondary metabolites in leaf and root tissues, and (iii) the combined effect of warming scenarios and *Pc* infection increases the susceptibility of *C. sativa* to *Pc*.

## 2. Results

### 2.1. Warming Scenarios in Chestnut

In non-infected plants, *C. sativa* seedlings showed no wilting or mortality. Plant growth was highest in non-infected seedlings exposed to high ambient temperature, and total fine root biomass was significantly higher in non-infected seedlings under high ambient temperature than under ambient temperature (Figure 1).

On day 0, photosynthetic rates (*P_n_*) and transpiration rates (*E*) were similar irrespective of the warming treatment (Figure 2A,B), whereas stomatal conductance (*g_s_*) values increased significantly in plants under high ambient temperature compared to plants under ambient temperature (*p* < 0.05; Figure 2C). On day 10, *P_n_*, *E* and *g_s_* values of seedlings under high ambient temperature and heat wave event scenarios were significantly lower than in seedlings under ambient temperature (*p* < 0.05; Figure 2). No differences were observed in plant growth, root biomass or leaf exchange parameters between families.

The phenolic compound profiling of seedlings differed depending on the warming scenario, mother tree and sampling time. Eleven of the 21 compounds identified in leaves and roots (Table 1) varied depending on the warming scenario (Table 2). More phenolic compounds showed changes in leaves than in roots (eight vs. three, respectively; Table 2).

Two compounds (quercetin 3-O-glucuronide and coniferyl aldehyde) varied significantly depending on the mother tree, and four compounds (ethyl gallate, ellagic acid acetyl-xyloside, procyanidin and hydroxytyrosol acetate) varied significantly depending on the sampling date (*p* < 0.05; Table 2). On day 0, the PCA plot clearly separated plants under ambient temperature from plants under high ambient temperature (Figure 3A). Disaggregation of groups occurred mainly throughout PC2 (Figure 3A), and the compounds that contributed significantly to this separation (in order of highest to lowest contribution) were catechin, hydroxytyrosol acetate, ellagic acid acetyl-xyloside and lariciresinol.

These four compounds were foliar and decreased with increasing temperature (Table 3). On the tenth day after seedlings from the warming scenarios had been exposed to ambient temperature (recovery), PCA revealed a clear separation between the three groups of seedlings (Figure 3B). The compounds that contributed significantly to disaggregation in PC1 were catechin and procyanidin in leaves, and those that contributed to disaggregation in PC2 were hydroxybenzoic acid, ellagic acid and coniferyl aldehyde in roots. With regard to ambient temperature, high ambient temperature frequently caused a decrease in phenolic compounds in plants, whereas heat wave events frequently caused an increase (Table 3).

### 2.2. Warming Scenarios and Phytophthora cinnamomi Infection in Chestnut

Ten days after inoculation, *P_n_*, *E* and *g_s_* parameters significantly decreased in infected plants previously exposed to ambient temperature (*p* < 0.05; Figure 2), whereas *P_n_*, *E* and *g_s_* remained unaltered in infected plants previously subjected to high ambient temperature and heat wave events. Thus, *Pc* was able to change the physiology of the plants in response to temperature by homogenising the values of gas exchange in leaves. At day 15, regardless of the warming treatment, most seedlings infected with *Pc* showed leaf chlorosis and wilting. Defoliation frequently occurred before plant death, which was faster and higher in seedlings exposed to ambient temperature than in seedlings under high ambient temperature (Figure 4). Thirty days after inoculation, the overall mortality of *Pc*-infected seedlings was 60.7%, and the final mortality rates of plants subjected to ambient temperature, high ambient temperature and heat wave events were 55, 60 and 69%, respectively. The two chestnut families used for phenolic compound profiling differed significantly in susceptibility to *Pc* (*p* < 0.05; Appendix A).

Four compounds of the secondary metabolism of plants showed significant or marginally significant changes in response to *Pc* (*p* < 0.05 and *p* < 0.10, respectively; Table 4). Ethyl gallate in leaves was significantly highest in *Pc*-infected seedlings (Figure 5A), and hydroxybenzoic acid, 4-hydroxyphenylacetic acid and coniferyl aldehyde in roots were lowest in *Pc*-infected seedlings (Figure 5B–D). However, ethyl gallate increased in plants exposed to ambient and high ambient temperatures but decreased in plants subjected to heat waves, 4-hydroxyphenylacetic acid decreased irrespective of the scenario, and coniferyl aldehyde decreased in plants exposed to ambient temperature but increased in plants subjected to heat waves (Figure 6). Eight of the 21 compounds identified in leaves and roots (Table 1) showed different changes depending on the warming scenario and *Pc* infection (significant *Pc* × scenario interaction in Table 4) (Figure 6). Miquelianin and hydroxytyrosol acetate showed different changes depending on the family and *Pc* infection (significant *Pc* × mother tree interaction in Table 4) (Appendix A).

At day 10, the PCA obtained from the compounds that varied in response to the warming treatments (Table 4) permitted visual differentiation between non-infected and *Pc*-infected plants (Figure 7). A clear separation between non-infected and *Pc*-infected plants was observed only in plants exposed to ambient temperature (Figure 7). The compounds that contributed most to the PC1 and PC2 axes were 3-feruloylquinic acid in leaves (81.3%); 4-hydroxyphenylacetic acid (8.7%), coniferyl aldehyde (3.9%) and hydroxybenzoic acid (3.6%) in roots; and ellagic acid in leaves (0.9%). The phenolic compound profile and variation in content were unique for each combination of warming scenario and *Pc* (Table 5).

## 3. Discussion

Climate change is increasing the Earth’s surface temperature and the frequency and intensity of extreme weather events such as heat waves. Moreover, climate change is altering the ways in which pathogens interact with plants and the ways in which plants resist pathogen attacks. This study provides, for the first time in *C. sativa*, information about the physiology and biochemistry of plants under altered warming scenarios of high ambient temperature and heat waves, and, for the first time in a tree, information on subsequent infection by the widespread pathogen *Pc*. The study provides relevant information that can be applied to the management of the species in the context of global change.

### 3.1. Effects of Temperature Increase in Chestnut

The increased growth and fine root biomass of *C. sativa* seedlings exposed to high ambient temperature are in agreement with studies of trees from temperate regions exposed to increasing temperature [51]. The root architecture is also influenced by increased temperature, although it can vary considerably between species [52]. Our increased-temperature scenarios did not significantly affect *P_n_* or *E* at the end of the 30-day period (day 0), although these parameters tended to increase in plants exposed to high ambient temperature (Figure 2). Warm temperatures in temperate regions may enhance the *P_n_* of trees by increasing tree photosynthetic pigments, depending on the species [53]. In general, *P_n_* reaches its maximum from 25 to 40 °C, but higher temperatures decrease *P_n_* due to impaired protein function [54]. After the 10-day recovery period, seedlings in the two warming scenarios had significantly lower *P_n_*, *E* and *g_s_* values than seedlings at ambient temperature. The causes of this decrease may have differed depending on the scenario. At high ambient temperature, plants may have undergone heat acclimation [55] by altering the optimum temperature at which *P_n_* was maximum. Conversely, seedlings subjected to the two heat waves could have had their PSII damaged and/or stress memory induced [56].

A decrease in total phenolic compounds has been reported in the leaves and roots of grapevine plants subjected to water, osmotic and cold stress [57,58,59]. Similarly, in *Quercus suber* seedlings exposed to low temperatures, a higher concentration of flavonoids was reported compared to control plants, suggesting that the biosynthesis of these compounds was encouraged by low temperatures [60]. In similar aged *C. sativa* seedlings subjected to heat stress, an increase in phenolic compounds was observed in roots [28], although the response varied depending on the tree origin. In trees, the biosynthesis of secondary metabolites in response to heat appears to be species- and compound-dependent [12]. In the present work, phenolic changes in chestnut in response to altered scenarios were genotype-dependent (Appendix A).

During recovery, seedlings previously exposed to high ambient temperature continued to show lower values of several phenolic compounds at day 10 than seedlings previously exposed to ambient temperature (Table 3). However, seedlings subjected to heat waves shifted from a decrease in phenolic compounds at day 0 to phenolic compound reestablishment, or an increase, at day 10. Delayed changes or shifts in trends in phenolic content occur when plants have entered a recovery phase after stress [57,59,61,62]. During the recovery phase, plants tend to restore phenolic content to ambient temperature levels, unless physiological damage occurs [28,63]. The increase in several phenolic compounds in our heat-wave-stressed plants may have occurred as a consequence of stress memory, as observed for *Alopecurus pratensis* [62], but this needs further study to be confirmed.

### 3.2. Phenolic Profile of Chestnut Seedlings after Infection by Pc

In studies on phenolic quantification in *C. sativa* plants infected by *Pc*, only Camisón et al. [10] used roots, the natural portal of entry for this pathogen. Nine days after the inoculation of susceptible *C. sativa* individuals with the same *Pc* strain used here, total phenolics decreased in roots and did not alter in leaves [10]. In the present study, we went a step further by obtaining a non-targeted phenolic profile of infected and non-infected plants, detecting significant variations in three compounds in response to *Pc* (one in leaves and two in roots; Figure 5). Phenolic compounds are involved in many plant defence mechanisms against abiotic and biotic stress [64,65]. In response to pathogens, they can act as constitutive defence compounds [13] or as elicitors and trigger systemic acquired resistance (SAR) for long-lasting immunity [66]. Ethyl gallate, which was significantly highest in the leaves of *Pc*-infected seedlings, acts as an antimicrobial agent and elicitor, and can activate the SAR pathway and induce pathogenesis-related (PR) resistance gene expression [66]. Hydroxybenzoic acid, better known as salicylic acid (SA), was significantly lowest in the roots of *Pc*-infected seedlings. SA is a stress-induced hormone and one of the key molecules in the signal transduction pathway involved in both local defence reactions at infection sites and the induction of SAR [67,68]. SA participates in the early stages of the defensive response of plants to infection [69], when *P. cinnamomi* acts as a biotrophic pathogen. Changes in ethyl gallate and hydroxybenzoic acid detected in our *Pc*-infected plants were probably involved in the mechanisms described above, but this needs to be tested.

Phenolic compounds also participate in the lignification of cell walls, an additional defensive response to pathogens [37]. Lignin enrichment of root cell walls has been reported as a plant-defensive response in the early stages (hours) after infection, although, in susceptible plants, it is not sufficient to stop pathogen hyphae from penetrating into the roots [70,71]. It has been reported that coniferyl aldehyde is used by plants for the hydroxylation of the enzyme ferulate-5-hydroxylase as a downstream substrate in the lignin pathway [72,73]. The lower levels of coniferyl aldehyde in *Pc*-infected plants may indicate that this compound was used for lignin biosynthesis. Lastly, 4-hydroxyphenylacetic acid has been reported to be metabolised by plant pathogens and used as a source of carbon and energy [74,75].

### 3.3. Enhanced Chestnut Resistance to Phytophthora cinnamomi after High Ambient Temperature

In *Pc*-infected plants, a high ambient temperature with subsequent *Pc* infection was the best combination for chestnut survival. This information may be useful for modelling when comparing different climate regions of the world. Based on climate change predictions for Mediterranean countries, the effects of temperature on the impact of *Phytophthora* species on trees have been addressed. In *Alnus glutinosa* in Northeast France, hot summers and cold winters ensure better tree survival after alder decline induced by *Phytophthora* x *alni*, probably because extreme temperatures limit the pathogen’s survival [76]. In Spain, increasing temperatures due to climate change are not expected to increase the impact of *Phytophthora* species on *Quercus ilex* acorn germination [77], although an increase in soil temperature of 1.5 °C may increase the inoculum of *Pc* in the soil [78]. In several ornamental plants in California, *P. ramorum* was most pathogenic at 20 °C and had intermediate pathogenicity at 25 °C and the lowest pathogenicity at 12 °C [79]. However, the increase in temperature is a single consequence of global change, among others, that could alter the susceptibility of chestnut trees to *Pc* [18]. In the Mediterranean area, the dieback of *Q. ilex* induced by *Pc* was exacerbated by severe drought [80,81]. Waterlogging in combination with subsequent water deprivation is the worst scenario for *Q. ilex* if soils are infested with *Pc* [82]. A higher frequency of extreme rain events that saturate the soil might be particularly beneficial for *Pc* (and negative for the tree [14]), potentially boosting its soil density beyond any possible defence response of the susceptible hosts [82,83]. However, an average, drier climate might imply suboptimal conditions for *Pc* infections, allowing for the slower advance of the disease in infested areas [83].

Why did seedlings exposed to high ambient temperature show lower and delayed mortality? Increased temperatures may have allowed high photosynthetic rates in plants, leading to the increased availability of soluble sugars (e.g., glucose and sucrose) for plant defence. However, non-structural carbohydrates were not assessed here, and measurements of *P_n_* performed on day 0 showed no evidence of differences in photosynthesis. In *Pc*-infested soil, seedlings at ambient temperature were the only group to show lower *P_n_*, *E* and *g_s_* values in response to *Pc* infection, in agreement with previous studies of susceptible *C. sativa* individuals [84,85]. In addition, seedlings exposed to ambient temperature showed a less developed root system than seedlings under high ambient temperature (Figure 1B), probably reducing their probability of survival, given that *Pc* zoospores infect and rot fine roots first [33]. It is also important to note that seedlings under high ambient temperature showed the greatest changes in phenolic content in response to *Pc*, in agreement with the more dynamic responses of hormones and metabolites in resistant than in susceptible chestnut clones [10]. The synthesis of heat shock proteins (HSPs) [86,87] further explains why seedlings exposed to high ambient temperature had lower and delayed mortality. Among the heat tolerance mechanisms in plants, the synthesis of HSPs occurs at temperatures close to that used by plants at a high ambient temperature [88]. HSPs also have an important role in defence signalling during pathogen attack, although some *Phytophthora* species, including *Pc*, are able to decrease HSP levels in the host by targeting their growth promoters [41]. Further studies are needed to determine whether high ambient temperature enhance HSP levels in chestnut at the time of inoculation and whether HSPs are decreased if plants are exposed to a warming scenario before *Pc* infection.

## 4. Materials and Methods

### 4.1. Plant Material

In October 2017, seeds were collected from four wild *C. sativa* trees growing in Central Spain (39°37′23.6′′ N, 5°23′1.6′′ W; 870 m a.s.l.; Castañar de Ibor, Las Villuercas, Extremadura). Approximately 200 nuts per tree were collected. The trees belonged to a group of 15 chestnuts more than 500 years old, known as Castaños de Calabazas, which are protected by regional legislation. They were chosen because giant, centuries-old trees are a reservoir of genetic diversity [89]. Trees were selected at least 70 m apart to minimise the chances of sampling intercrossed individuals.

Immediately after collection, seeds were immersed in a fungicide solution (2 g L^−1^ Thiram 80GD, ADAMA Inc., Spain) for 5 min and those that floated were discarded as non-viable. Viable seeds were washed in sterile water and stored at 4 °C for two weeks.

In November 2017, viable seeds were individually weighed and sown at 1 cm depth in 48-cell plastic root trainers (one seed per cell). Individual cells were 330 mL in volume, 18 cm high, 5.3 × 5.3 cm upper surface and contained peat (PKN1 Florava^®^ Peat Substrate, pH 5–6). Germination was assessed weekly and plants were kept in natural light conditions under shade (50% solar radiation) and watered to field capacity until they were well established. The greenhouse was located at the Faculty of Forestry of Plasencia (40°02′ N, 6°04′ W; 374 m a.s.l., Extremadura region, Spain).

### 4.2. Treatments and Experimental Design

Two greenhouses and one climate chamber were used to assess the secondary metabolism of chestnut in response to warming and *Pc* scenarios. At the end of May 2018, when seedlings were six months old, they were divided into three groups and the following treatments were applied: (i) ambient temperature, (ii) high ambient temperature and (iii) ambient temperature and two separate heat waves. Treatments lasted 30 days (Figure 8). The first group was placed in a greenhouse where the temperature from 11.00 a.m. to 5.00 p.m. was set at 30 °C (ambient temperature), simulating the midday temperature of a meso-Mediterranean bioclimate zone in June. According to the IPCC’s fifth assessment, the global average air temperature is expected to rise by 0.3 to 4.8 °C by the end of this century [90]. Based on this forecast, the second group was placed in a greenhouse where the temperature from 11.00 a.m. to 5.00 p.m. was set at 35 °C (high ambient temperature). The greenhouses were similar in terms of size and sun exposure (50% solar radiation). The third group was placed in the first greenhouse for 15 days (30 °C 11.00 a.m. to 5.00 p.m.), moved to a climate chamber for 3 days (45 °C 11.00 a.m. to 5.00 p.m.; first heat wave), returned to the first greenhouse for 9 days (30 °C 11.00 a.m. to 5.00 p.m.) and then placed in the climate chamber for 3 days (45 °C 11.00 a.m. to 5.00 p.m.; second heat wave). This treatment simulated two recent identical heat waves in Europe [91]. The climate chamber had translucent walls and received 50% solar radiation. Irrespective of treatments, plants were well watered and the volumetric soil water content (VWC) was kept at values close to 30%, optimal for the correct growth of chestnut [25]. VWC was verified using a TDR 100 soil moisture meter (Spectrum Technologies Inc., Plainfield, Illinois, USA). The relative humidity conditions were the same in the three scenarios. At the end of June 2018, 30 days after treatments started (corresponding to the end of the second heat wave), half of the seedlings were exposed to ambient temperature to assess recovery (non-infected plants, Figure 8) and half were inoculated with *Pc* (infected plants, Figure 8).

Plants were arranged following a split-plot random design replicated in three blocks, with the warming scenarios as the main factor (3 categories: ambient temperature, high ambient temperature and heat wave events; whole plots) and the mother trees as the split factor (4 categories). Two root trainers per block and warming scenario were used. In the three blocks, the mother trees were represented in each whole plot by 24 individuals from the four open-pollinated families. Individuals were randomly positioned within the blocks. The experiment comprised 864 plants corresponding to 3 blocks × 3 warming scenarios × 4 families × 24 individuals. For *Pc* inoculation, one root trainer per block and warming scenario was used.

### 4.3. Inoculation of Phytophthora cinnamomi and Mortality Assessment

A single A2 strain of *Pc* (coded Ps-1683) isolated from a diseased *C. sativa* tree in Galicia (43°18′32′′ N, 8°13′57′′ W, Northern Spain) was used. The strain was proven to be highly virulent in *C. sativa* [10,35,92]. *Pc* inoculum was prepared following Jung et al. [93] and incubated at 20–25 °C in total darkness for four weeks. Soil was infested by mixing 12 mL inoculum with the first three cm of substrate in each individual cell, taking care not to damage the roots of the seedlings. To promote the better establishment of the pathogen, approximately 50 g freshly formed *C. sativa* leaves were mixed with the substrate and inoculum in each 48-cell plastic root trainer. After inoculation, seedlings were irrigated, left for one day and then flooded for two days with non-chlorinated water to promote sporangia production and zoospore release. Symptoms and mortality induced by *Pc* were monitored for 30 days. In August 2018, to fulfil Koch’s postulates, *Pc* was successfully re-isolated on PARPH selective medium from the roots of infected seedlings.

### 4.4. Plant Measurements and Sampling

Measurements and sampling for biochemical analysis were conducted on the last day of warming treatments (day 0) and 10 days after these treatments had ceased (Figure 8). These sampling dates corresponded to day 0 (pre-inoculation) and day 10 after infection, respectively. The effects of treatments and *Pc* infection on plants were evaluated by (i) leaf symptoms, plant growth, plant biomass and plant mortality; (ii) leaf gas exchange; and (iii) quantification of phenolic compounds. At the end of the experiment (day 30), plant growth and biomass were evaluated only in non-infected plants.

External symptoms were evaluated by the visual characterisation of leaf discoloration observed in seedlings. Plant growth was expressed as the difference in seedling height before and after warming treatments (30-day period).

Leaf gas exchange parameters (net photosynthetic rate (*P_n_*), transpiration rate (*E*) and stomatal conductance (*g_s_*) were determined at days 0 and 10 in 10–15 seedlings per scenario and treatment (Figure 8). Leaf gas exchange parameters of seedlings were measured using a portable differential infrared gas analyser (IRGA; Li-6400, Li-Cor Inc., Lincoln, NE, USA) connected to a broadleaf chamber. Measurements were taken from 9 a.m. to 2 p.m., with temperatures of 28–30 °C, 33–35 °C and 42–45 °C in the ambient temperature, high ambient temperature and heat wave event scenarios, respectively, and photosynthetically active radiation (PAR) ranging from 500 to 800 μmol photons m^−2^ s^−1^ (daylight and variable PAR conditions). In the recovery and inoculation phase, temperatures ranged from 28 to 30 °C in all seedling groups.

At days 0 and 10, leaves and roots of six seedlings per treatment and scenario (from two mother trees) were sampled to obtain the non-targeted phenolic compound profiling of plants. Destructive sampling was carried out by removing the seedling from the alveolus, collecting the leaves and carefully separating the peat from the roots. Leaves and roots were dried at room temperature in complete darkness using silica gel. Once the samples were dry, they were ground to a fine powder in a ball mill (Mixer Mill MM 400, Retsch, Germany) to pass through a 0.42 mm screen.

Plant biomass was assessed by destructive sampling of 32 seedlings per warming scenario (8 seedlings per family), separated by organs (leaf, stem, fine root and coarse root) and dried in an oven at 60 °C to a constant weight on a precision scale.

### 4.5. Non-Targeted Phenolic Compound Profiling

Two of the four chestnut families, selected at random, were used. Fine leaf and root powder (0.3 g and 1 g, respectively) was homogenised with 70% ethanol (5 mL and 8 mL, respectively) and sonicated for 15 min. The extracts were cold-macerated at 4 °C for 24 h and centrifuged at 3500 rpm for 15 min, and the supernatant was filtered using a 0.45 µm filter. The filtered extracts were stored at −80 °C until analysis.

The analysis, identification and quantification of phenolic compounds were performed with an LC-MS system (HPLC 1260-QTOF 6550, Agilent Technologies, Santa Clara, CA, USA). A total of 20 µL filtered extract of each sample was injected onto a Spherisorb C 18 (250 × 4.6 mm ID, 5 µ) reversed-phase column at a rate of 1 mL/min. The mobile phase comprised a gradient elution (water with 2% formic acid in methanol) from 1% water to 100% methanol of 100 min for leaves and 50 min for roots. Chromatograms were recorded at a wavelength of 350 nm for flavonoids and 280 nm for phenols. Concentrations of the compounds were estimated from a standard curve (0.00, 0.01, 0.05, 0.10, 0.20 mg/mL) using gallic acid, ellagic acid, quercetin or quercetin 3-O-rutinoside, catechin or procyanidin. The results were expressed in mg of equivalents per g of dry weight.

### 4.6. Statistical Analysis

To evaluate the effect of the warming treatments on the leaf gas exchange, growth and biomass of *C. sativa* seedlings, a one-way analysis of variance (ANOVA) was performed using “warming scenario” as a single factor. To analyse the combined response of seedlings to the warming treatments and *Pc*, a two-way ANOVA was performed, including “warming scenario” and “presence of *Pc*” as main effects, and their interaction. To identify significant differences between means, Tukey’s multiple comparison tests were used at *p* < 0.05.

General linear models (GLM) that included the amount of a particular compound as a dependent variable were used to detect the effect of warming scenarios and/or *Pc* on the phenolic compound profiles of *C. sativa* plants. “Warming scenario” and/or “presence of Pc”, “mother tree” and their interactions were used as fixed factors, and “seed weight”, “time to emerge” and “plant height” as covariates. To estimate the variation in the time of phenolic compounds in response to warming scenarios and/or *Pc*, the fixed factor “time” was added to the models. The residuals of the models were tested for normality, and means were compared using Tukey’s HSD test.

To obtain patterns of variation in the significant compounds detected in the GLMs, three principal component analyses (PCA) were performed: for warming scenarios at day 0, warming scenarios at day 10 and warming and *Pc* infection scenarios together. PCAs were performed using the R statistical package ‘factoextra’. Before the PCAs, data were centred and standardised to reduce scale effects.

Survival time analysis based on the Kaplan–Meier estimate was used to analyse the time to death of infected seedlings exposed to different warming scenarios and compare survival probabilities to *Pc* [94].

Before analyses, data were checked for normality and homogeneity of variances. ANOVAs, GLMs and Tukey’s tests were performed with STATISTICA v10 software [95].

## 5. Conclusions

The following conclusions can be drawn:Chestnut seedlings exposed to high ambient temperature (35 °C) showed the highest vigour in plant growth, fine root biomass and dynamic response of phenolic compounds to biotic stress. Plant mortality induced by *Pc* was 20% lower in chestnuts previously exposed to high ambient temperature (for 30 days) than in chestnuts previously exposed to ambient temperature. This result is encouraging for the future persistence of chestnut in the Mediterranean area, where temperatures are increasing and the presence of *Pc* is becoming more frequent.Two 45 °C heat waves for three days did not alter plant growth, fine root biomass or chestnut’s susceptibility to *Pc*. This suggests the good adaptation of chestnut to heat waves in the absence of water limitation.*Pc* was able to alter the physiology of *C. sativa* plants in response to temperature by homogenising the values of gas exchange parameters in leaves.In response to heat, changes in the phenolic compound profiles of chestnut plants exposed to high ambient temperature and heat waves were similar. However, during recovery, most phenolic compounds of plants exposed to high ambient temperature remained low, but, in plants subjected to heat waves, they increased. Changes in compounds were greater in leaves than in roots.Three phenolic compounds (ethyl gallate in leaves, 4-hydroxyphenylacetic acid in roots and coniferyl aldehyde in roots) showed significant variations in chestnut in response to *Pc* infection. Five additional phenolic compounds showed different changes in their content in response to *Pc* and the scenario that plants were exposed to before inoculation.Variation was observed in plasticity at the family level of several phenolic compounds in response to altered warming scenarios. This variation would be an opportunity for *C. sativa* to respond and probably adapt to global warming.

## Figures and Tables

**Figure 1 plants-12-00556-f001:**
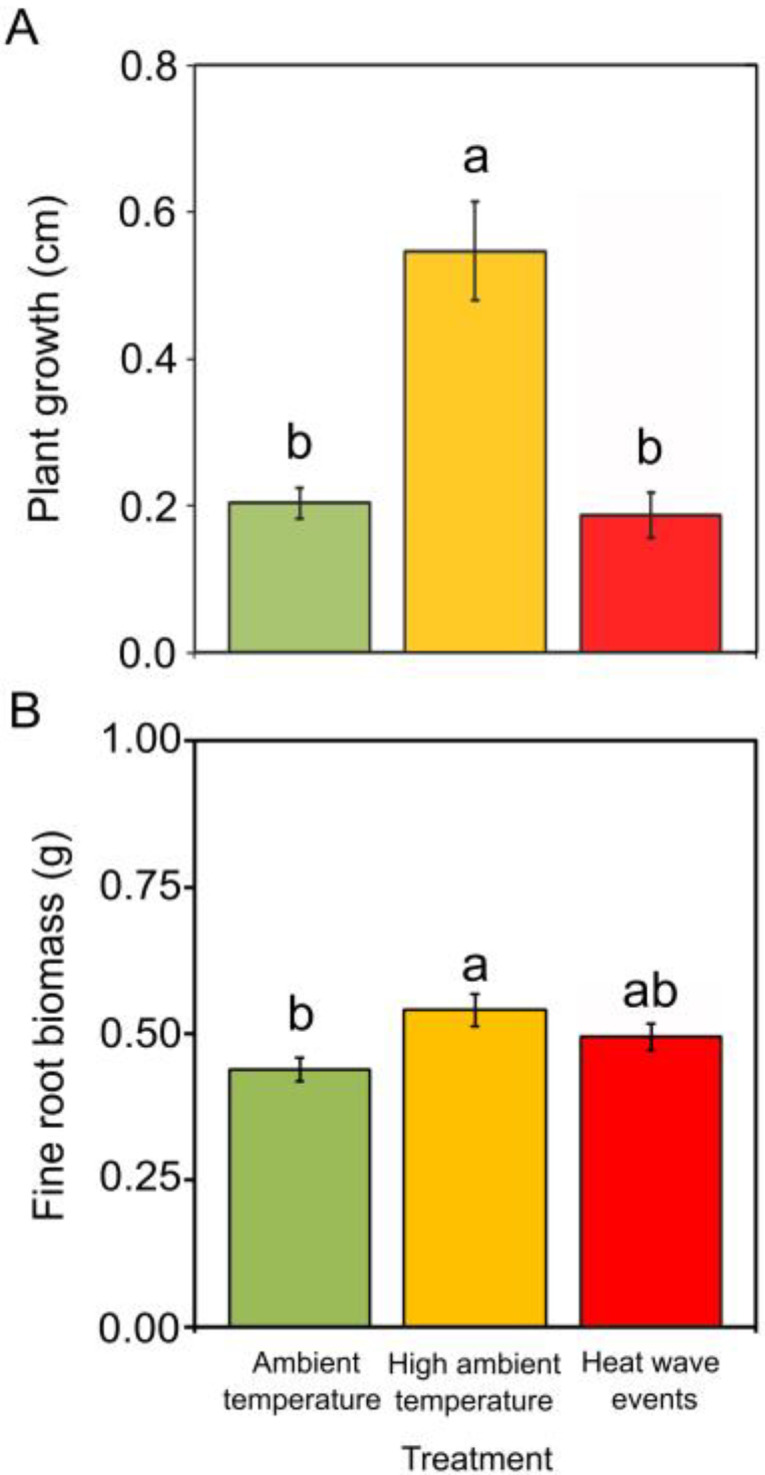
Plant growth (**A**) and fine root biomass (**B**) of six-month-old *Castanea sativa* seedlings exposed to ambient temperature (green), high ambient temperature (orange) and two heat waves (red). Vertical bars are standard errors, and different letters indicate significant differences between means (Tukey’s HSD tests at *p* < 0.05).

**Figure 2 plants-12-00556-f002:**
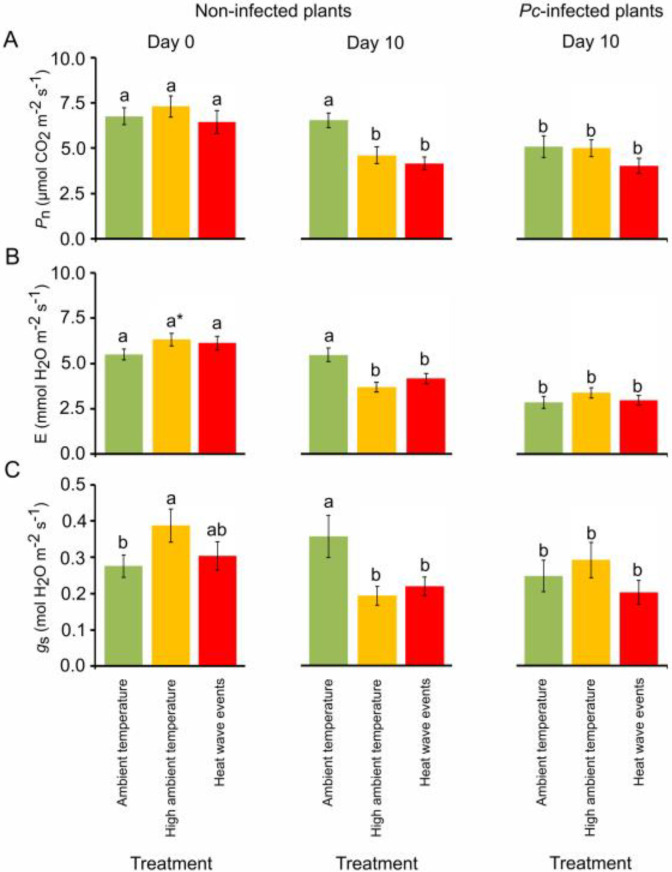
Photosynthetic rate values (*P_n_*) (**A**), transpiration rates (*E*) (**B**) and stomatal conductance (*g_s_*) (**C**) of *Castanea sativa* seedlings exposed to ambient temperature (green), high ambient temperature (orange) and two heat waves (red). Measurements were obtained 0 and 10 days after plants were exposed to treatments (non-infected plants) and 10 days after plants were exposed to treatments and *Phytophthora cinnamomi* (*Pc*) infection (*Pc*-infected plants). Vertical bars are standard errors, different letters indicate significant differences between means (*p* < 0.05), and asterisks indicate marginally significant differences (*p* < 0.10) (Tukey’s HSD tests).

**Figure 3 plants-12-00556-f003:**
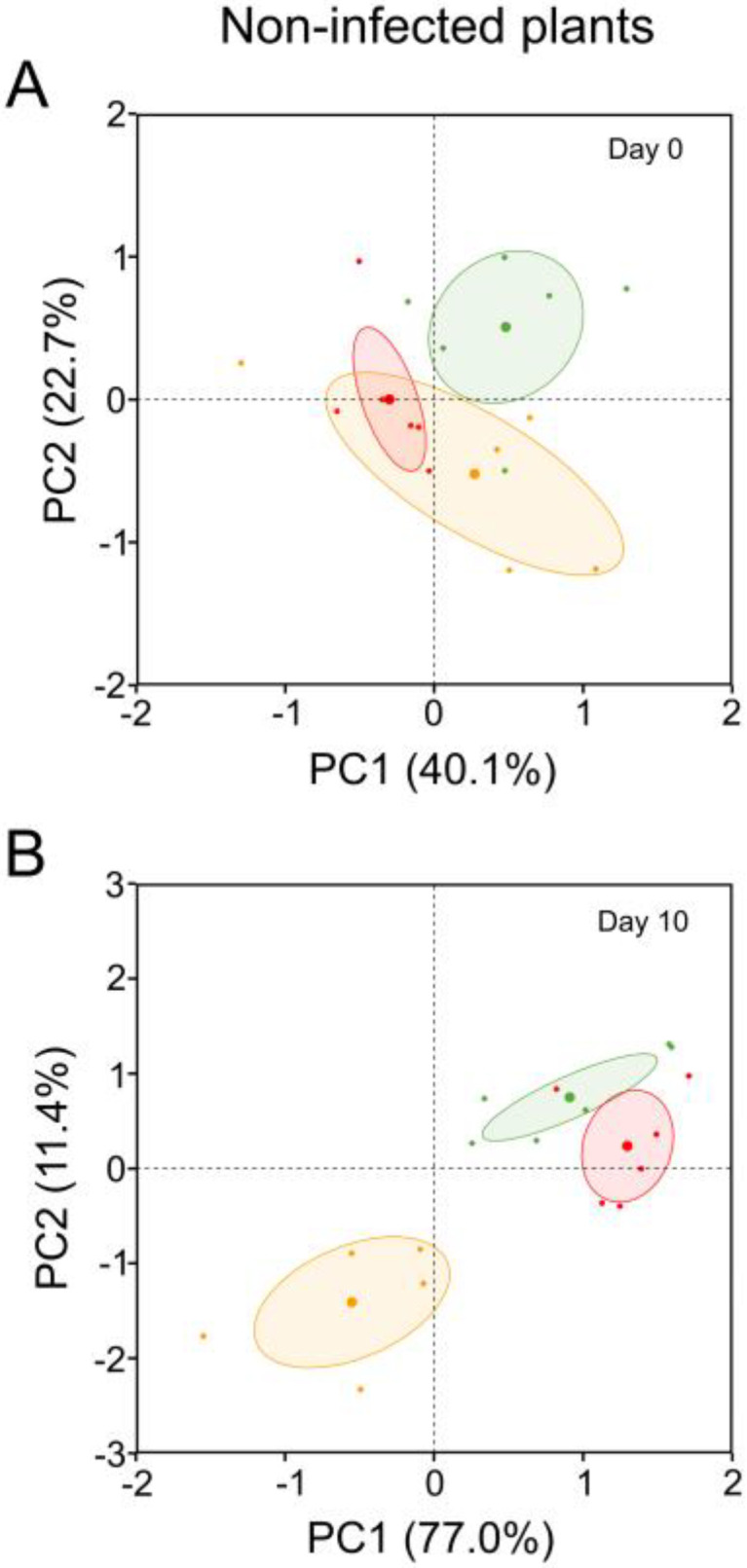
PCA of phenolic compounds included in Table 2 of non-infected *Castanea sativa* seedlings exposed to ambient temperature (green circles), high ambient temperature (orange circles) and two heat waves (red circles).

**Figure 4 plants-12-00556-f004:**
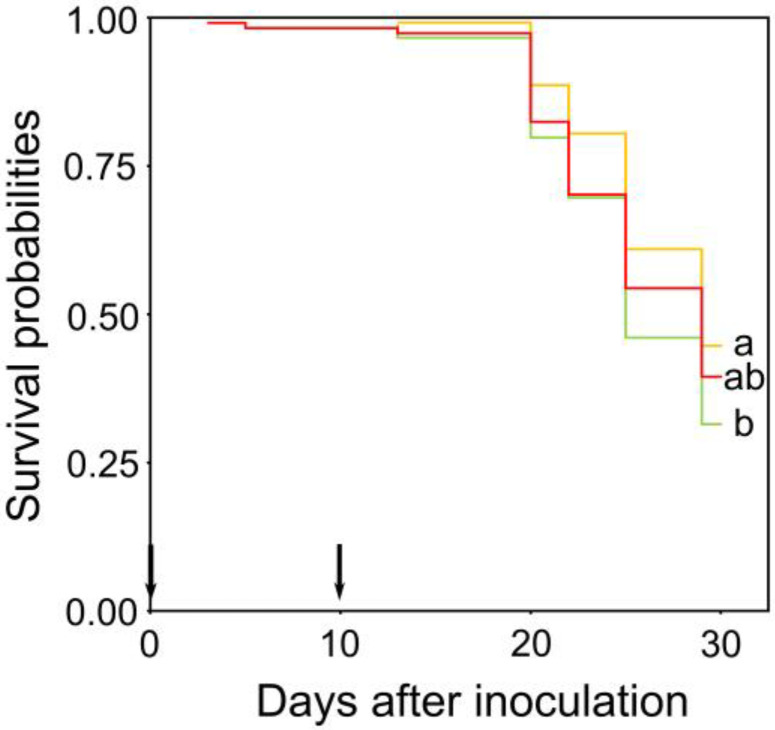
Survival probabilities of *Castanea sativa* seedlings exposed to ambient temperature (green), high ambient temperature (orange) and heat waves (red) and infected at day 0 with *Phytophthora cinnamomi*. Global log-rank test was significant at *p* = 0.080. Different letters indicate significant differences between survival curves (*p* < 0.05) and arrows indicate the dates of phenol assessment.

**Figure 5 plants-12-00556-f005:**
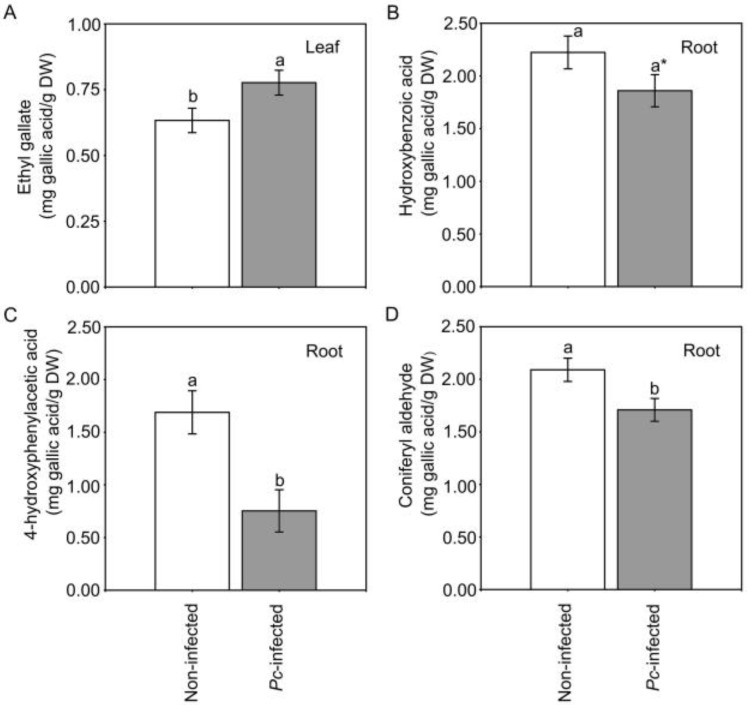
Mean values of phenolic compounds ethyl gallate (**A**), hydroxybenzoic acid (**B**), 4-hydroxyphenylacetic acid (**C**), and coniferyl aldehyde (**D**) of *Castanea sativa* seedlings not infected or infected with *Phytophthora cinnamomi* (*Pc*). According to the linear mixed models shown in Table 4, the four compounds were significantly affected by *Pc* infection. Measurements were taken 10 days after inoculation. Vertical bars are standard errors, different letters indicate significant differences (*p* < 0.05), and the asterisk indicates a marginally significant difference (*p* < 0.10) (Tukey’s HSD tests).

**Figure 6 plants-12-00556-f006:**
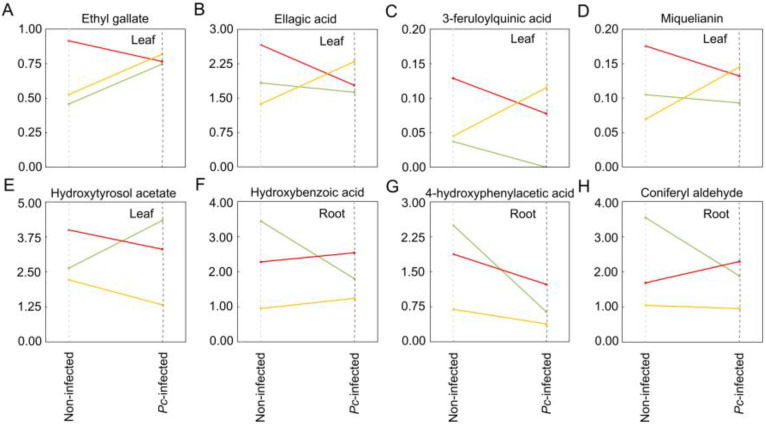
Phenolic compounds of *Castanea sativa* plants that showed different changes in their content in response to *Phytophthora cinnamomi* infection and the scenario plants were exposed to before inoculation (significant *Pc* × scenario interactions in Table 4; *p* < 0.05). Phenolic content was obtained at day 10 after inoculation in plants previously exposed to ambient temperature (green lines), high ambient temperature (orange lines) and two heat waves (red lines).

**Figure 7 plants-12-00556-f007:**
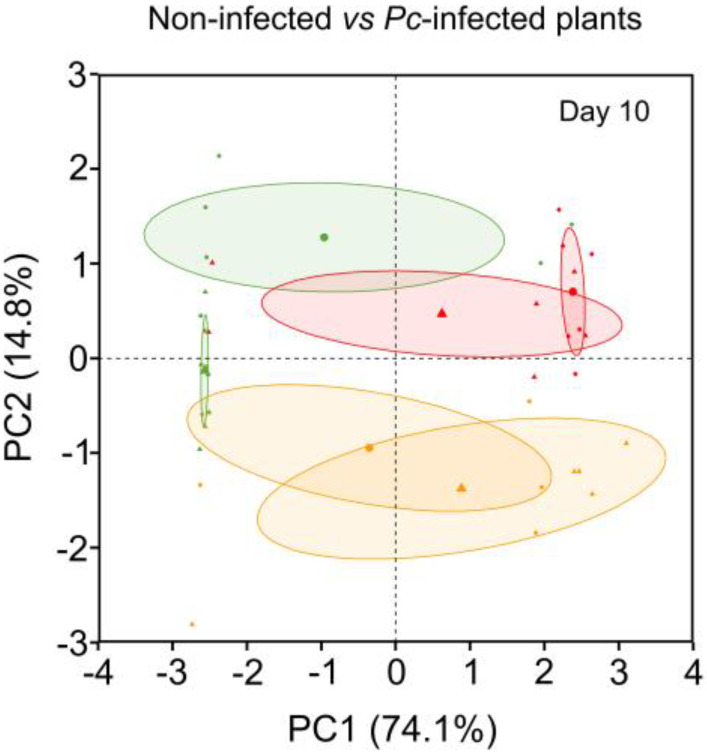
PCA of phenolic compounds included in Table 4 of non-infected *Castanea sativa* seedlings exposed to ambient temperature (green circles), high ambient temperature (orange circles) and two heat waves (red circles) (non-infected plants) and *Phytophthora cinnamomi*-infected *C. sativa* seedlings previously exposed to ambient temperature (green triangles), high ambient temperature (orange triangles) and two heat waves (red triangles) (*Pc*-infected plants). The five compounds that most contributed to the PC1 and PC2 axes were leaf 3-feruloylquinic acid (81.3%), root 4-hydroxyphenylacetic acid (8.7%), root coniferyl aldehyde (3.9%), root hydroxybenzoic acid (3.6%) and leaf ellagic acid (0.9%).

**Figure 8 plants-12-00556-f008:**
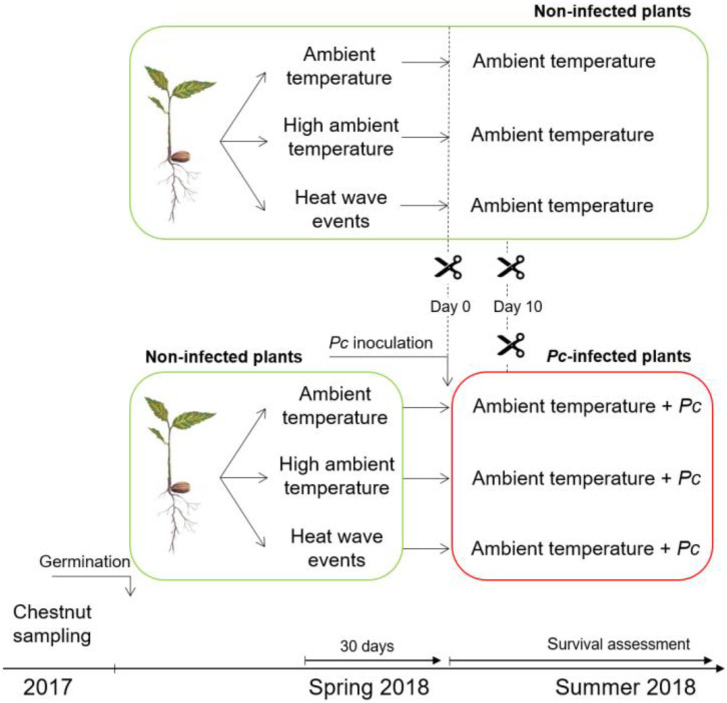
Experimental design and sampling plan.

**Table 1 plants-12-00556-t001:** Secondary metabolite compounds detected, identified and quantified in leaves and roots of six-month-old *Castanea sativa* plants.

Class	Subclass	Leaf	Root
Phenols	Hydroxybenzoic acids	Ethyl gallate	Hydroxybenzoic acid
		Ellagic acid acetyl-xyloside	Gallic acid
		Ellagic acid	Ellagic acid
	Hydroxycinnamic acid	3-feruloylquinic acid	
	Hydroxyphenylacetic acid		4-hydroxyphenylacetic acid
Lignans		Lariciresinol	
Flavonoids	Flavanols	Procyanidin	
		Catechin	
	Flavonols	Miquelianin (quercetin 3-O-glucuronide)	Quercetin 3-O-galactoside
		Quercetin 3-O-rutinoside	Kaempferol-3-O-(6″ acetyl) glucoside 7-O rhamnoside
		Quercetin 3-O-rhamnoside	
		Quercetin 3-O-galactoside	
Other polyphenols		Hydroxytyrosol	Tyrosol
		Hydroxytyrosol acetate	Coniferyl aldehyde (4-hydroxy-3-methoxycinnamaldehyde)

**Table 2 plants-12-00556-t002:** Results of general linear models for analyses of chemical changes in seedlings of two *Castanea sativa* mother trees in response to warming scenarios. Significant *p*-values are indicated in bold (*p* < 0.05), asterisks indicate marginally significant differences (*p* < 0.10), and ns indicates non-significant result.

		Leaf								Root		
Effects	Df	Ethyl Gallate	Ellagic Acid Acetyl-xyloside	Ellagic Acid	Lariciresinol	Procyanidin	Catechin	Quercetin 3-O-glucuronide	Hydroxytyrosol Acetate	Ellagic Acid	Hydroxybenzoic Acid	Coniferyl Aldehyde
Scenario [S]	2	**0.012**	**0.004**	**0.006**	**0.019**	**0.001**	**0.002**	**0.009**	**0.039**	**0.015**	**<0.001**	**<0.001**
Mother tree [M]	1	*	ns	ns	ns	ns	ns	**0.034**	ns	ns	ns	**0.007**
Time [T]	1	**0.012**	**0.022**	ns	ns	**0.035**	ns	ns	**0.014**	ns	ns	ns
S × M	2	*	ns	ns	ns	ns	ns	**0.007**	ns	ns	ns	ns
S × T	2	**0.029**	**0.037**	*	ns	**0.006**	ns	**<0.001**	**0.015**	**0.002**	**0.002**	**0.022**
M × T	1	*	ns	ns	ns	**0.046**	ns	ns	ns	*	ns	ns
S × M × T	2	ns	ns	ns	ns	ns	ns	ns	ns	**0.005**	**0.005**	**0.022**
Seed weight (g)	1	**0.028**	ns	ns	ns	**0.014**	ns	ns	ns	ns	ns	ns
Time to emerge (d)	1	ns	ns	ns	ns	ns	ns	**0.017**	**0.017**	ns	ns	ns
Plant height (cm)	1	**0.021**	ns	ns	ns	ns	ns	ns	ns	ns	ns	ns

**Table 3 plants-12-00556-t003:** Changes in phenolic compounds in leaves and roots of *Castanea sativa* seedlings 0 and 10 days after exposure of plants to high ambient temperature and heat wave events compared to plants exposed to ambient temperature. Red and blue indicate increase and decrease, respectively. Two arrows indicate significant variation (*p* < 0.05), one arrow indicates marginally significant variation (*p* < 0.10), and ns indicates non-significant variation (Tukey’s HSD test).

		Non-Infected Plants			
		Day 0		Day 10	
Organ	Compound	High Ambient Temperature	Heat Wave Events	High Ambient Temperature	Heat Wave Events
**Leaf**	Ethyl gallate (mg gallic acid/g DW)	ns	ns	ns	**↑↑**
	Ellagic acid acetyl-xyloside (mg gallic acid/g DW)	**↓↓**	**↓↓**	**↓↓**	ns
	Lariciresinol (mg gallic acid/g DW)	**↓**	**↓**	**↓↓**	ns
	Procyanidin (mg procyanidin/g DW)	ns	ns	**↓↓**	**↑↑**
	Catechin (mg catechin/g DW)	**↓↓**	ns	**↓↓**	ns
	Miquelianin (quercetin 3-O-glucuronide) (mg quercetin/g DW)	ns	ns	ns	**↑↑**
	Hydroxytyrosol acetate (mg gallic acid/g DW)	**↓↓**	**↓↓**	ns	**↑**
	Ellagic acid (mg ellagic acid/g DW)	ns	ns	ns	**↑↑**
**Root**	Ellagic acid (mg ellagic acid/g DW)	ns	ns	↑↑	ns
	Hydroxybenzoic acid (mg gallic acid/g DW)	ns	ns	**↓↓**	**↓↓**
	Coniferyl aldehyde (mg gallic acid/g DW)	ns	ns	**↓↓**	ns

**Table 4 plants-12-00556-t004:** Results of general linear models for analyses of chemical changes in seedlings of two *Castanea sativa* mother trees in response to warming scenarios and *Phytophthora cinnamomi* infection. Significant *p*-values are indicated in bold (*p* < 0.05), asterisks indicate marginally significant differences (*p* < 0.10), and ns indicates non-significant result.

		Leaf					Root		
Effect	Df	Ethyl Gallate	Ellagic Acid	3-feruloylquinic Acid	Miquelianin	Hydroxytyrosol Acetate	Hydroxybenzoic Acid	4-hydroxyphenylacetic Acid	Coniferyl Aldehyde
*Phytophthora* [*Pc*]	1	**0.032**	ns	ns	ns	ns	*	**0.006**	**0.014**
Scenario [S]	2	**0.018**	ns	**0.038**	**0.015**	**<0.001**	**<0.001**	**0.009**	**<0.001**
Mother tree [M]	1	ns	ns	ns	*	ns	ns	ns	**0.031**
*Pc* × S	2	**0.016**	**0.014**	**0.044**	**0.005**	**<0.001**	**0.003**	*	**<0.001**
*Pc* × M	1	ns	ns	ns	**0.037**	**0.015**	ns	ns	ns
S × M	2	ns	ns	ns	ns	**<0.001**	ns	ns	ns
*Pc* × S × M	2	ns	ns	ns	ns	ns	**0.031**	ns	*
Seed weight (g)	1	ns	ns	ns	ns	ns	ns	ns	ns
Time to emerge (d)	1	ns	ns	*	ns	**0.002**	ns	ns	ns
Plant height (cm)	1	ns	ns	ns	ns	ns	ns	ns	*

**Table 5 plants-12-00556-t005:** Changes in phenolic compounds in leaves and roots of *Phytophthora cinnamomi*-infected *Castanea sativa* seedlings 0 and 10 days after exposure of plants to high ambient temperature and heat wave events compared to plants exposed to ambient temperature. Red and blue indicate increase and decrease, respectively. Two arrows indicate significant variation (*p* < 0.05), one arrow indicates marginally significant variation (*p* < 0.10), and ns indicates non-significant variation (Tukey’s HSD test).

		*Pc*-Infected Plants (Day 10)	
Organ	Compound	High Ambient Temperature + Pc	Heat Wave Events+ Pc
**Leaf**	Ethyl gallate (mg gallic acid/g DW)	ns	ns
	Ellagic acid (mg ellagic acid/g DW)	ns	ns
	3-Feruloylquinic acid (mg gallic acid/g DW)	↑↑	↑
	Miquelianin (quercetin 3-O-glucuronide) (mg quercetin/g DW)	↑↑	ns
	Hydroxytyrosol acetate (mg gallic acid/g DW)	↓↓	↓↓
**Root**	Hydroxybenzoic acid (mg gallic acid/g DW)	ns	↑
	4-Hydroxyphenylacetic acid (mg gallic acid/g DW)	ns	ns
	Coniferyl aldehyde (mg gallic acid/g DW)	↓↓	ns

## Data Availability

Data are contained within the article.

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
