# Peer review of "Warming Scenarios and Phytophthora cinnamomi Infection in Chestnut (Castanea sativa Mill.)"

_plants, 2023, doi:10.3390/plants12030556_

Round 1
Reviewer 1 Report
Dear authors,
Below are my recommendations:
1. For every significant results mentioned in text, in the results section, the p Value for that particular analysis should be included, in brackets. For example on line 85, put the p value for the analysis of fine biomass in brackets.
2. The conclusion is poorly written. You basically repeated sections of the conclusion. Write a proper conclusion, preferably in paragraphs form, pointing out what you deduce from the findings of the study.
Author Response
Dear reviewer, the authors would like to thank you for your time and helpful comments, which have helped to improve the manuscript.
- For every significant results mentioned in text, in the results section, the p Value for that particular analysis should be included, in brackets. For example on line 85, put the p value for the analysis of fine biomass in brackets
You were right. The p-values have been added in most sentences throughout the results section.
- The conclusion is poorly written. You basically repeated sections of the conclusion. Write a proper conclusion, preferably in paragraphs form, pointing out what you deduce from the findings of the study.
We welcome your suggestion and you are right to point out that we have repeated sections of the discussion for the conclusions section. However, this is our style. When addressing the discussion, we like to discuss a particular result or conclusion. Merging the conclusions into a paragraph will not allow to differentiate the different types of concusions, referring to non-infected plants, infected plants, physiology, bichemistry and adaptation. Since all these concepts are broad and different in nature we prefer to mantain the conclusions by numbers. In fact, reviewer 3 congratulated us for spellng the conclusions in this way. In order to follow your suggestion we added a couple of sentences with the deductions of our work.
Thank you again, and kind regards, the authors
Reviewer 2 Report
The manuscript “Warming Scenarios and Phytophthora cinnamomi Infection in Chestnut (Castanea sativa Mill.) presents a well-structured experimental design to evaluate the impacts of high temperature/weat waves and Phytophthora cinnamomi infection on Castanea sativa plants.
The manuscript presents a complete review of the topic, with clear results and discussion.
Specific comments:
Line 100 and line 167 - Only count 21 compounds identified in leaves and roots (table 1).
Line 111. Table 1. It would be advantageous to adjust table 1 in order to facilitate reading. Also, Quercetin 3-O-glucuronide and 4-hydroxy-3-methoxycinnamaldehyde are referred in the text, tables and figures by the respective synonyms Miquelianin and Coniferyl aldehyde, so synonyms should be included in table 1.
Line 136 – has a repeated period.
Line 291-292 – Phytophthora x alni (as the x represents a complex of species should not be in italic).
Ensure that article 28 is properly published before being included in the references.
28. Dorado, F.J.; Pinto, G.C.; Monteiro, P.; Chaves, N.; Alías, J.C.; Rodrigo, S.; Camisón, Á.; Solla, A. Heat stress and recovery effects on the physiology and biochemistry of Castanea sativa Mill. Front. For. Glob. Chang. 2023, In press. doi: 10.3389/ffgc.2022.1072661
Author Response
Dear reviewer, the authors would like to thank you for your time and very helpful and kind comments.
Line 100 and line 167 - Only count 21 compounds identified in leaves and roots (table 1).
We greatly appreciate the alert to this error which had gone completely overlooked. You are absolutely right. The sentences have been changed with 21 compounds instead of 23.
- Line 111. Table 1. It would be advantageous to adjust table 1 in order to facilitate reading. Also, Quercetin 3-O-glucuronide and 4-hydroxy-3-methoxycinnamaldehyde are referred in the text, tables and figures by the respective synonyms Miquelianin and Coniferyl aldehyde, so synonyms should be included in table 1.
Done
- Line 136 – has a repeated period.
Done
- Line 291-292 – Phytophthora x alni (as the x represents a complex of species should not be in italic).
Done
- Ensure that article 28 is properly published before being included in the references. [28. Dorado, F.J.; Pinto, G.C.; Monteiro, P.; Chaves, N.; Alías, J.C.; Rodrigo, S.; Camisón, Á.; Solla, A. Heat stress and recovery effects on the physiology and biochemistry of Castanea sativa Front. For. Glob. Chang. 2023, In press. doi: 10.3389/ffgc.2022.1072661].
The article was recently published. It can be found at the following link:
https://www.frontiersin.org/articles/10.3389/ffgc.2022.1072661/full
The reference has been updated
Reviewer 3 Report
The authors have reported an interesting timely study linking increasing temperature and the effects that abiotic factor will have on an important biotic factor, Phytophthora cinnamomi. As the authors have stated, very few studies have combined these important agents of change in the health of forests. The manuscript is well written and although the interactions between increased temperature and its effects on both pathogen and host are complex the authors have done a good job, particularly in the conclusions of spelling out the most important findings.
The authors could tie their findings more closely to other climate change forest disease forecasts and outline how their mixed results differ from other more dire scenarios of how diseases will in many instances likely increase.
The authors do need to be consistent in referring to the inoculated and un-inoculated seedlings. Referring to these same treatments as epidemic and non-epidemic is not correct.
I have made a number of relatively minor edit suggestions in the attached pdf.

Author Response
Dear reviewer, thank you so much for your time, big effort and helpful comments, which have helped to improve the manuscript a lot.
- The authors could tie their findings more closely to other climate change forest disease forecasts and outline how their mixed results differ from other more dire scenarios of how diseases will in many instances likely increase.
In the discussion we added several sentences, and 5 new references, trying to do address this. However, we were cautious and focussed on the effects of climated change and Pc on trees only. There is a huge amount of literature to add about climate change and forest disease forecasts, wich could be summarized in a review paper.
Line 81: I see the authors have followed the mdpi order of sections with results and discussion coming before materials and methods. MDPI should make sure their instructions to authors are consistent as it is not clear. The instructions state both the order the authors have followed and the traditional order where materials and methods follows the introduction. both
We have followed the template structure provided by the journal for the submission of articles. Initially, we had the standard structure of a research article, which we have adapted to the template in terms of structure and the order in which figures are mentioned.
- Lines 96; 143-145: 'isn't one of the more interesting findings in Fig 2 that the epidemic treatment appeared to change the response of the plants to temperature? Ambient temp led to more photosynthesis, transpiration and stomatal conductance in the absence of Pc but when Pc was inoculated the ambient temp was no longer any different that high temps?'. 'This is an important finding as mentioned earlier, maybe it should be moved to the paragraph that refers to Fig 2 earlier. The story would be easier to follow if this paragraph referring to Fig 2 was closer to Fig 2 in the text'.
Thanks a lot for telling. We added an additional conclusion for that, but we preferred not to move the sentences. Please note that the first and second sections of the results refer to non-infected and Pc-infected plants, respectively.
- Lines 108; 149; 186; 286: 'rather than calling the diseased treatment 'epidemic' would it not be better to refer to the treatment as infected and uninfected? Epidemic typically refers to the extent of infection (a very high incidence of infection) but the treatment could be better expressed as simply infected vs uninfected'.
We fully agree and have therefore decided to use the terms non-infected and Pc-infected plants instead of non-epidemic and epidemic scenarios. We have made all appropriate changes throughout the text, in tables and figures.
- Lines 213-217: 'need a clearer opening paragraph that points to the most significant implications of this study. This first paragraph is more like the objective stated in the Introduction. Need to put this study into broader context of uncertainty of disease and host response to a warming climate associated heat waves etc'.
Done
- Line 339: 'very nice figure! would it be a little clearer if the text-Epidemic scenarios- was moved to above the red box, and the red box extended to the left to be consistent with the non-epidemic scenario above?'
Thank you very much, we are glad you liked the figure. We have moved the text above the red box. However, we have not extended the red circle into the left side since these plants were not infected.
ESPECIFIC COMMENTS
- Line 11: suggest including common name (chestnut) here as it ties directly to the paper title
Done
- Line 15: 'a heat wave event' or 'heat wave events'
Heat wave events. Done
- Line 22: ' waves'?
Waves. Done
- Line 29: 'To the best of our knowledge, this is the first study address the combined...'
We agree. The change was done.
- Line 50: ' considerably'
We agree. The change was done
- Line 55: suggest 'little' rather than 'no'
We agree. The change was done
- Line 58: 'of'
Done
- Line 67: suggest spelling 'P. cinnamoni' rather than starting a sentence with abbreviation
We agree. The change was done
- Line 67: 'a large' enormous is too dramatic.
We agree. The change was done
- Line 77: 'events'
Done
- Line 92: suggest spelling these out first, Photosynthetic rate and transpiration rate
We agree. The changes were done.
- Line 93: 'stomatal conductance'
Done
- Line 99: 'differed'
Done
- Line 101: 'varied'
Done
- Line 117: which 2 compounds? state which ones
The names of the compounds were added.
- Line 117: 'varied significantly'
Done
- Line 172: 'varied'
Done
- Line 223: 'increased temperature scenarios'
Done
- Line 239: 'similar aged?'
We agree. The change was done
- Line 257: 'this pathogen', some pathogens enter through leaves, others through bark.
You are absolutely right. We have added your very good suggestion which we had overlooked.
- Line 306: ', probably reducing their probability of survival'.
We agree. The change was done
- Line 308: 'changes'
Done
- Line 318: 'decreased'
Done
- Line 324: ' are'
Done